# Differential BACH1 Expression in Basal-like Breast Tumors of Black Women Identified via Immunohistochemistry

**DOI:** 10.3390/curroncol32070404

**Published:** 2025-07-14

**Authors:** N. M. Dowling, Galina Khramtsova, Olufunmilayo Olopade, Shabnam Samankan, Bok-Soon Lee, Jiyoung Lee

**Affiliations:** 1Department of Acute & Chronic Care, School of Nursing, Ashburn, VA 20147, USA; mndownling@gwu.edu; 2Department of Epidemiology, Milken Institute School of Public Health, Washington, DC 20037, USA; 3Center for Aging, Health and Humanities, George Washington University, Washington, DC 20006, USA; 4Center for Clinical Cancer Genetics and Global Health, Department of Medicine, University of Chicago, Chicago, IL 60637, USA; galina@uchicago.edu (G.K.);; 5GW Medical Faculty Associates, George Washington University Hospital, Washington, DC 20037, USA; shsamankan@mfa.gwu.edu; 6Department of Biochemistry and Molecular Medicine, School of Medicine and Health Sciences, George Washington University, Washington, DC 20037, USA; bslee2021@gwu.edu; 7GW Cancer Center, George Washington University, Washington, DC 20037, USA

**Keywords:** BACH1, MCT1, IHC score, Black women, race, disparity, breast cancer, basal-like subtype

## Abstract

BACH1 mRNA has been known to promote breast cancer metastasis and predict the survival of breast cancer patients. We analyzed BACH1 protein levels in tumor tissues using immunohistochemistry assays. Expression levels of transcription factor BACH1 differ in tumors from Black women compared to those from White women. When we used a patient cohort of 130, higher levels of BACH1 were linked to tumors that were larger in size and a specific type of breast cancer called basal-like breast cancer. Another protein, MCT1, which is connected to BACH1, was also positively linked in tumors from Black women. These findings suggest that functional molecule BACH1 proteins are highly expressed in tumors from Black women, as well as in the basal-like subtype of breast tumors.

## 1. Introduction

Despite improvements in overall survival rates and patient outcomes, breast cancer is the second-highest cause of cancer-related death in women [1]. Breast cancer-related death is mainly due to cancer metastasis, which results in secondary breast tumor formation in distant organs such as the brain, liver, lungs, and bones. The metastatic process consists of multiple steps, including epithelial to mesenchymal transition (EMT), migration and invasion, intravasation, extravasation, and micro-metastasis [2]. Additionally, tumor microenvironmental features, including hypoxia and tumor-infiltrating immune cell populations, contribute to the cancer metastatic processes [3]. A heme-binding transcription factor, BTB and CNC homology 1 (BACH1), has been shown to promote breast cancer metastasis, particularly in triple-negative breast cancer (TNBC), which is one of the most aggressive and lethal subtypes of breast cancer because of the lack of targeted therapy [4,5,6,7]. Physiological functional roles of BACH1 include intracellular heme homeostasis, redox regulation, ferroptosis, and metabolic regulation in cancer cells [8,9,10]. In breast cancer, relatively high levels of BACH1 cause increased migration and invasion, intravasation, and in vivo metastasis by upregulating the transcriptional expression of matrix metalloproteinase 9 (MMP9) and C-X-C chemokine receptor type 4 (CXCR4) [11]. The depletion of BACH1 expression in cancer cells was sufficient to reduce the migration, invasion, intravasation, and metastasis of breast cancer. BACH1 also promotes EMT transition by downregulating epithelial cell markers through forkhead box 1 (FOXA1) inhibition in pancreatic cancer cells [12]. In lung cancer, metastasis rates are increased by upregulated metabolic pathways, such as hexokinase 2 (HK2) and glyceraldehyde-3-phosphate dehydrogenase (GAPDH), in a BACH1-dependent manner, which is further stabilized by an antioxidant treatment or activated nuclear factor-erythroid factor 2-related factor (NRF2) (equivalent to the mutation of Kelch-like ECH associated protein 1 (Keap1)) [13,14]. In many tumor types, including breast tumors, BACH1 regulates transcriptomes to facilitate cancer cell migration, invasion, and metastasis. Therefore, *BACH1* mRNA expression has been associated with poor prognosis in breast cancer patients, particularly patients with basal-like TNBC tumors [6,13,14]. Yet, it is unclear whether BACH1 expression levels are differentiated by tumor characteristics or the clinical variables of patients.

Previously, we revealed that BACH1 regulates metabolic pathways, including lactate catabolism in TNBC cells, by suppressing monocarboxylate transporter 1 (MCT1 or encoded by *SLC16A1*), which facilitates pyruvate and lactate transport via a proton-dependent mechanism [15,16,17]. MCT1 expression is highly associated with recurrent breast-invasive ductal carcinoma and TNBC, as well as the metastasis of melanoma [18,19,20]. Mechanistically, MCT1 supports antioxidant regulation required for the successful completion of the metastatic processes of melanoma cells. Targeting MCT1-dependent lactate flux contributes to improved outcomes for cancer patients, as MCT1 is associated with tumorigenesis. Currently, a small molecule targeting MCT1, AZD3965, is in a Phase I clinical trial as an anti-cancer therapeutic [21,22], indicating MCT1 as a valuable biomarker for breast cancer.

In our study, we evaluated the expression levels of BACH1 or MCT1 in breast tumor tissues to identify them as useful biomarkers. Furthermore, the expression levels of BACH1 or MCT1 were analyzed using the following clinical variables: breast cancer subtypes, tumor stages, tumor grade, and tumor size. Moreover, the expression levels of BACH1 or MCT1 were analyzed using demographic variables, such as patient age and race, to identify biological risk factors in our patient data cohort, which includes both Black and White patients from the United States. Our patient tumor cohort is highly valuable since it contains a similar number of patients of each race to investigate BACH1 and MCT1 expression in breast tumors.

## 2. Materials and Methods

### 2.1. Patient Tumor Collection and Tissue Microarray Construction

Archival formalin-fixed and paraffin-embedded (FFPE) breast tumor tissues were obtained from the Human Tissue Resource Center (HTRC) of the University of Chicago for tissue microarray (TMA) construction and were approved by the local Institutional Review Board (IRB # 10760B) at the University of Chicago. Patient data were collected from 1999 through 2001 and used for our study. Informed consent was obtained from all subjects and their guardians. The TMAs from FFPE in situ, invasive carcinoma tumor samples, and adjacent histologically normal epithelium tissues served as internal positive controls. Tissue cores of 1 mm were arrayed into a new recipient paraffin block using an Automated Tissue Arrayer (ATA-27, Beecher Instruments, Sun Prairie, WI, USA), as described previously [23]. The location and identification of each tissue core were recorded in a Microsoft Excel database. After tumor tissues were fixed using 10% formaldehyde, dehydrated using 70% ethanol, and embedded into a paraffin block, they were cut into sections 3 μm in thickness before staining. Pathologic features, including diagnosis, grade, tumor size, and axillary lymph node metastasis, were abstracted from pathology reports. The histology diagnosis, or grading of invasive breast cancer and carcinoma in situ, was performed separately by a pathologist (G.K.) based on the protocols of the College of American Pathologists and the World Health Organization (WHO) Classification Protocol for the examination of specimens from patients with invasive carcinoma of the breast [24]. Breast cancer subtypes were defined as luminal A (estrogen receptor (ER)^+^ and/or progesterone receptor (PR)^+^, human epidermal growth receptor HER2^−^), luminal B (ER^+^ and/or PR^+^, HER2^+^), basal-like [(ER^−^, PR^−^, HER2^−^, CK5/6^+^) and/or epidermal growth factor receptor (EGFR)^+^], HER2^+^ (HER2^+^, ER^−^, PR^−^), or unclassified (negative for all five markers), as described previously [25].

### 2.2. Immunohistochemistry

The TMAs contain a total of 130 patient tumors and were used for BACH1 and MCT1 staining via immunohistochemistry assays, which were performed in Dr. Olopade’s lab and the HTRC at the University of Chicago. Briefly, the TMA slides were processed via heat-induced antigen retrieval (HIAR) in Trish-EDTA (pH 9) buffer using steamer and treated with antibodies against BACH1 (Santa Cruz #c-271211, Dallas, TX, USA) or MCT1 (Millipore Sigma #SAB2702323, Bedford, MA, USA) in a 1:25–1:50 dilution in TE buffer (pH 9.5) overnight at 4 °C. After being counter-stained using hematoxylin (Agilent #CS700, Santa Clara, CA, USA), slides were scanned and digitized on a 3DHISTECH Pannoramic^TM^ whole-slide scanner and further analyzed by G.K. For the IHC control groups, colon, lung, and tonsil tissue sections were adapted as a positive control, and the xenografted tumor sections using BACH1 or MCT1-knock down human breast cancer cells were used as a negative control.

### 2.3. TMA Scoring

The TMA slides stained for BACH1 and MCT1 were digitized using the 3D HISTECH panoramic whole-slide scanner and analyzed. Scoring was based on intensity and the percentage of positively stained cells; all discrepancies were resolved by a second examination using a digital slide image. The data corresponds to the results from the Human Protein Atlas (https://www.proteinatlas.org/URL accessed on 6 May 2021). Scoring was confirmed using light microscopy and was performed independently and semi-quantitatively by experienced pathologists (G.K. and S.S.) and one researcher (J.L.). Tissues that failed IHC staining and scoring were eliminated from future analysis. For MCT1 staining, both intensity and percentage were scored for H-score calculation, as MCT1 is a membrane protein [26,27]. The intensity of protein expression was recorded as follows: 0 (no staining), 1 (weak staining, light brown), 2 (moderate staining, brown), or 3 (strong staining, dark brown). The proportion of tumor cells was scored as follows: 0 (<10% positive cells), 1 (10–20% positive cells), 2 (21–50% positive cells), or 3 (>50% positive cells).

For H-scores, the percentage of cells at each staining intensity level was calculated, and finally, an H-score was assigned using the following formula [28,29]:
[1 × (% cells 1+) + 2 × (% cells 2+) + 3 × (% cells 3+)]

The final score, ranging from 0 to 300, assigns a greater relative weight to higher-intensity membrane staining in each tumor sample. The sample can then be considered positive or negative based on a specific discriminatory threshold.

NOTES: Originally, a score of less than 50 was considered negative (−), and scores of between 50 and 100 were considered weakly positive (+1). However, many centers have lowered the threshold, and only cases scoring less than 10 are considered negative, with those scoring between 10 and 100 being weakly positive.

For BACH1 staining, both intensity and proportion were scored using the Allred scoring system, as BACH1 is a transcription factor [30]. The Allred score combines the percentage of positive cells and the intensity of the reaction product in most carcinomas. The two scores (intensity and proportion) are added together to yield a final score with 8 possible values. Scores of 0 and 2 are considered negative. Scores of 3–8 are considered positive. For the analysis, we converted one score to another to streamline the results [30,31].

### 2.4. Statistical Analyses

For co-expression analyses using cancer patient data, we used nonparametric (distribution-free) approaches. The Mann–Whitney U test (Wilcoxon rank-sum test) was used to compare outcomes between two independent groups. To compare three or more groups on a dependent variable we used the Kruskal–Wallis’s test. For a significant Kruskal–Wallis’s omnibus test, we conducted post hoc pairwise comparisons using the Dunn test [32]. *p*-values for post hoc pairwise comparisons were adjusted using the Holm method to reduce type I errors [33]. Proportional odds ordinal logistic regression (PO-OLR) was used to control for multiple covariates. We used PO-OLR as a generalization of the Kruskal–Wallis test that extends to multiple covariates and interactions [34]. Spearman’s rank order correlation coefficient was employed to test the strength and magnitude of the relationship between ranked variables of interest. To assess the relationship between a dichotomous categorical variable and an ordinal variable, we used the rank biserial correlation. Scatterplots and distribution plots were used to study the associations between the variables of interest in the sample and examine outliers and unusual observations. In particular, we estimated differences in BACH1 and the target gene MCT1 expression among multiple patient groups and clinical variables, such as race (Black vs. White), age (below 55 vs. 55 and older), tumor subtypes (luminal A, luminal B, basal-like, and HER2-positive), tissue types (ductal carcinoma in situ, lobular carcinoma in situ, hyperplasia, lymph node metastasis, and tumors), tumor size (diameter), tumor grades (grade 1: well differentiated, grade 2: moderately differentiated, grade 3: poorly differentiated, and ND: not determined), and invasiveness (metastatic vs. non-metastatic). All hypothesis tests were two-sided and carried out at an alpha level set at 0.05. Statistical analyses were conducted in *R* version 4.3.1 (R Core Team, Vienna, Austria, 2023). The R package *rms*, Version 6.7-0 was used to fit the PO-OLR models [35].

## 3. Results

### 3.1. Detection of BACH1 and MCT1 in the Breast Tumor Tissues Using IHC Analysis

The transcriptional regulatory factor BACH1 activates or suppresses its target gene expression. As BACH1 mRNA levels predict breast cancer patient’s outcomes, BACH1 protein levels have potential as a biomarker to stratify cancer patients [6,7,10,11,13,14]. Thus, we immediately determined whether BACH1 protein expression levels were accessible via IHC analysis using the breast tumor tissues. The TMA blocks contained tumor tissue samples collected from 130 patients with all subtypes (basal, luminal A, luminal B, and HER2-positive) or tissue types (ductal carcinoma in situ breast cancer (DCIS) and lobular carcinoma in situ (LCIS)) at all stages, as well as normal tissues (Table A1). Available clinical information included tumor histology, tumor grade, tumor stage, tumor size, patient’s race, and patient’s age at diagnosis. Patients under age 55 at diagnosis comprised 40% (*N =* 52/130) of the total. The racial composition was 54.6% Black (*N* = 71/130), 40.7% White (*N =* 53/130), and 4.6% Asian or Hispanic (*N =* 6/130). The third group was excluded from the race-related analyses due to the small sample size. The number of patients, by descriptive measures, is indicated in Table A1. For scoring IHC staining, BACH1 IHC was evaluated using Allred score totals ranging from 0 to 7, and MCT1 IHC was scored using an H score total of 3 × 2 × 1×, ranging from 0 to 295. The median, mean, and standard deviation (SD) of IHC scores by clinical parameters are summarized in Table A2. Using IHC assays, we detected BACH1 expression in breast tumors collected and processed in our facility (Figure 1A).

### 3.2. BACH1 Expression Levels Are Positively Associated with Breast Tumor Size

Using BACH1 IHC scores, we analyzed the association between BACH1 levels and the biological variables of patient tumors. Given our relatively small sample size for subgroups of interest and non-normal distributions of BACH1 scores, we used nonparametric models for the statistical analyses in our study [30]. Our general hypothesis was to determine whether BACH1 levels were differentiated according to tumor characteristics. We tested BACH1 expression levels using IHC scores by tumor size, measured by diameter. An expression correlation analysis indicated that BACH1 IHC scores were positively correlated with tumor size. That is, BACH1 protein expression levels were higher in tumors with bigger diameters (*Spearman* coefficient = 0.207, *p* = 0.027) (Figure 1B). Likewise, we further divided patient tumors into two groups based on tumor diameter, specifically, small tumors (3–25 mm in diameter, *N* = 65) vs. big tumors (27–85 mm in diameter, *N* = 50) (Appendix A). This split corresponded to the 50th percentile of the tumor size distribution. The mean value of BACH1 IHC scores in smaller tumors was 3.292 (*SD* = 1.1693), whereas it was 4.2 (SD = 1.796) in bigger tumors. When BACH1 IHC scores were compared in the two tumor diameter groups using the Mann–Whitney U test for independent samples, we found a significant difference (*p* = 0.015) between the groups, showing that the bigger tumor group had higher BACH1 IHC scores than the smaller tumor group (Figure 1C).

### 3.3. BACH1 Expression Differs by Tumor Grade, Not by Tumor Tissue Type

We examined whether BACH1 levels differed by histological tumor grade, categorized as follows: Grade 1: well-differentiated, Grade 2: moderately and intermediate differentiated, Grade 3: poorly differentiated and dedifferentiated, and ND: cell type not determined or unclassified. A Kruskal–Wallis’s rank sum test rejected our omnibus hypothesis that each tumor grade had the same BACH1 expression level based on histological tumor grade. BACH1 levels were different in at least one grade (*p* = 0.015) (Appendix A). Based on the post hoc pairwise comparisons, BACH1 expression levels were significantly higher in the Grade 1 tumor group than the Grade 3 group (*p* = 0.0488). BACH1 expression levels were also different between the Grade 3 tumor group and the ND group (*p* = 0.009), with higher expression levels in the Grade 3 group compared to the ND group, which is an unclassified group.

We also analyzed BACH1 IHC scores by tumor tissue type, and our data showed similar levels of BACH1 for the following tumor tissue types: DCIS, LCIS, hyperplasia, lymph node metastasis (LN-MTS), and tumors (T) (Appendix A). Metastases (MTS) and LCIS were excluded from this comparison due to the limited sample size (*N* = 1 per group). Taken together, our data showed that BACH1 protein expression levels were relatively abundant in the Grade 3 tumor group, which was defined as poorly differentiated or dedifferentiated in our patient cohort.

### 3.4. BACH1 Expression Is Higher in Basal-like Breast Tumors than in the Other Subtypes

We further assessed whether BACH1 expression levels differed by the breast tumor subtypes. Prior research found that *BACH1* mRNA expression levels were highest in the basal-like subtype of breast tumors [6,10]. To investigate BACH1 score comparison, as well as other tests presented in our analyses, we used Holm *p*-value adjustments when the omnibus hypothesis for the Kruskal–Wallis nonparametric test was rejected. We also used alpha < 0.01 as the threshold, instead of alpha < 0.05, to minimize the type I error rate. The mean value of BACH1 scores, 4.906 $\left(SD=1.748;\;N=32\right),$ in the basal-like tumors is substantially higher than the mean values of 3.303 $\left(SD=1.558;\;N=76\right)\;$ in the luminal A tumors, 2.875 (SD=1.458; N=8) in the HER2-positive tumors, and 2 (SD=1.673; N=6) in the luminal B subtypes, according to a Mann–Whitney U test (Figure 2A). An independent group comparison of basal-like (*N* = 32) and luminal A (*N* = 76) subtypes, which have the most abundant patient samples, indicates that BACH1 IHC scores were significantly higher in the basal-like tumor subtype (*p* < 0.001, Rank-Biserial Correlation = 0.517) (Figure 2B). Our analyses indicate that the basal-like subtype has the highest levels of BACH1 expression compared to the other subtypes.

Since BACH1 levels were higher in the basal-like tumor subtype, we investigated whether BACH1 expression levels differed by tumor invasiveness or between metastatic and non-metastatic tumors. Tumors were divided into two groups: metastatic vs. non-metastatic tumors. The invasive tumor group (*N* = 15), which we refer to as metastatic, contained cancer types of MTS and LN_MTS; the non-metastatic group (*N* = 109) contained cancer types of T, DCIS, and hyperplasia. Unexpectedly, the results revealed no significant differences in BACH1 levels between the metastatic and non-metastatic tumor groups using the Wilcoxon W test (Appendix AA,B).

### 3.5. BACH1 Expression Is Higher in Tumors from Black Women than Those from White Women, Particularly in the Basal-like Subtype

In the United States, Black women have approximately 40% higher mortality rates from breast cancer than White women, although Black women have a lower incidence rate than White women [31]. This cancer disparity requires immediate clinical and scientific attention to identify which patients are at higher risk and what factors contribute to this disparity. Furthermore, precise and reliable biomarkers are needed to predict outcomes based on patients’ biological traits, including race. We analyzed BACH1 expression levels by race using the tumors from Black (*N* = 69) and White (*N* = 49) patients (Table A2). The mean value of BACH1 IHC scores for White women was 3.02 ± 1.942 (mean ± SD), and it was 3.971 ± 1.514 (mean ± SD) for Black women (Figure 3A). Analyses using the Mann–Whitney U test indicated that tumors from Black women showed significantly higher BACH1 scores than tumors from White women (*p* = 0.014, VS-MPR is 6.076). In addition, the total count of samples was higher for the tumors from Black women, with higher BACH1 scores than tumors from White women (Appendix A). We further dissected subtypes of tumors among Black and White women to investigate whether BACH1 levels differed by tumor subtype and race. The mean value of BACH1 IHC scores was markedly higher in the basal-like subtype than other subtypes of tumors among Black women, whereas the mean BACH1 scores were quite similar regardless of tumor subtype among White women (Figure 3B). Because race is an important indicator for the prognosis of breast cancer patients, we further investigated associations between BACH1 and race, controlling for tumor grade and tissue type. We estimated a PO-OLR to evaluate whether race affected BACH1 expression, while controlling for tissue type (metastatic vs. non-metastatic) and histological grades (tumor grade) [34]. We found that tumors from Black women had significantly higher BACH1 expression levels compared to those from White women when tumor grades were used as a controlled variable (*p* = 0.0399). In addition, when tissue type (metastatic or non-metastatic) was used as a controlling variable, tumors from Black women expressed significantly higher BACH1 levels than those from White women (*p* = 0.014, VS-MPR is 6.076), indicating different levels of BACH1 by race, despite tissue type or tumor grade. Taken together, BACH1 is expressed significantly more in tumors from Black patients than in those from White patients, and it is particularly highest in the basal-like tumors from Black women, regardless of tissue type or tumor grade.

### 3.6. BACH1 Expression Shows No Correlation with Patient Age

Age is a known risk factor for many cancers, including breast cancer, with breast cancer incidence increasing with patient age. Therefore, we examined whether BACH1 expression levels in breast tumors correlate with patients’ age in our cohort. Patients in our analyses were classified into two age groups: a younger group (below 55 years old, *N* = 50) and an older group (above 55 years old, *N* = 73), as most women experience menopause between the ages of 45 and 58. BACH1 IHC scores in both groups were compared using the Mann–Whitney U test. The mean BACH1 IHC score for tumors from the younger age group were 3.78 (SD = 1.866), with coefficient variation (0.494), while the mean was 3.521 (SD = 1.725), with coefficient variation (0.490), in the older age group. BACH1 expression levels were not significantly different in the two groups (*p* = 0.257) (Appendix AA). We further separated patients into three age groups: between 24 and 49 years old (*N* = 41), which is the representative group before menopause, between 50 and 65 years old (*N* = 39), and between 66 and 96 years old (*N* = 44). We compared BACH1 scores across multiple groups using the Kruskal–Wallis’s test. A statistical difference was not detected (*p* = 0.191) in BACH1 expression across the three age groups, indicating that BACH1 expression levels do not significantly differ by patient age (Appendix AB).

### 3.7. MCT1 Expression Levels Are Higher in Basal-like Tumors Regardless of Patient Race

Upregulated MCT1 levels in breast cancer, especially in basal-like subtype tumors, have previously been associated with poor outcomes for patients with breast cancer (18–20). We validated MCT1 expression by breast cancer subtypes using IHC assays in our patient cohort and analyzed its levels by patient race. Our TMA included MCT1 IHC staining scores (*N* = 120) from both Black women (*N* = 68) and White women (*N* = 52) (Table A2). The MCT1 IHC scores (Hscoretotal 3 × 2 × 1×) ranged from 10 to 295, with a mean of 171.4 (SD = 105.9), for tumors from Black women, and ranged from 0 to 295, with a mean of 140 (SD = 104.14), for tumors from White women. Representative IHC images for MCT1 staining, as shown as a membrane protein, are displayed with a wide range of staining scores, from low intensity to strong intensity (Figure 4A, Appendix A). Consistent with previous reports using IHC assays, we validated higher MCT1 expression in basal-like tumors when compared with the HER2^+^, luminal A, or luminal B subtype tumors (Figure 4B) [18,19,20]. The two most abundant subtypes, basal-like (*N* = 34) and luminal A (*N* = 78), were compared in terms of MCT1 expression levels. The mean value of MCT1 IHC scores for basal-like tumors was 221.765 (SD = 94.716), and it was 126.282 (SD = 97.119) for the luminal A subtype; their difference was significant according to the Mann–Whitney U test (*p* < 0.0001) (Figure 4C). For the Mann–Whitney U test, an effect size of 0.561 was given by the rank biserial correlation. Additionally, we investigated whether MCT1 expression showed racial disparity as BACH1 had shown. When MCT1 expression levels were analyzed by tumor subtype in each race, MCT1 scores were markedly different in the basal-like subtypes in both Black and White women (Figure 4D). Since MCT1 expression showed enrichment in the basal-like subtypes in both races, we further investigated whether MCT1 expression was equal or differed by patient race. The analysis of MCT1 IHC scores indicates no difference in MCT1 levels between Black and White women (*p* = 0.081 according to a Mann–Whitney U test) (Figure 4E). For the Mann–Whitney U test, an effect size of –0.286 is given by the rank biserial correlation. Taken together, these findings demonstrate that MCT1 expression was substantially elevated in the basal-like subtype of breast tumors compared to other subtypes, regardless of patient race.

### 3.8. MCT1 Expression Differs in the Histological Grade 3 Tumor Group

Next, we explored whether MCT1 levels were different or equal in terms of histological tumor grade, as we did for the BACH1 analysis. We approached this question with the same hypothesis, which stated that each tumor grade had the same MCT1 expression level; however, the Kruskal–Wallis’s rank sum test result indicated that the MCT1 expression levels differed. Post hoc comparisons indicated that MCT1 levels were significantly higher in the Grade 3 tumor subgroup than the Grade 2 subgroup (*p* = 0.004) in our patient cohort (Figure 5).

### 3.9. MCT1 Expression Has No Association with Tissue Types, Tumor Size, or Patient Age

We further investigated whether MCT1 expression differed by tumor tissue type and tumor size. MCT1 expression did not differ by either tumor tissue type (Appendix A) or size (Appendix A). Since MCT1 is also abundant in basal-like tumors, which are the most invasive subtype of breast cancer, we questioned whether MCT1 expression levels differed by tumor invasiveness. Tumors were divided into invasive (or metastatic tumors) (*N* = 20) and non-invasive (equivalent to non-metastatic tumors) (*N* = 105) groups and compared using the Wilcoxon W test (Appendix A). This analysis revealed no significant difference in MCT1 levels between the invasive and non-invasive tumor subgroups (*p* = 0.808). Moreover, we assessed whether MCT1 expression levels differed by patient age. Comparisons of the MCT1 IHC scores of younger (below 55 years old, *N* = 52) and older (above 55 years old, *N* = 72) groups returned no significant differences (Mann–Whitney U test, *p* = 0.179) (Appendix A). Likewise, comparing three age groups (24–49 years old, 50–65 years old, and 66–96 years old) revealed no statistical differences in the MCT1 IHC scores (Kruskal–Wallis test, *p* = 0.191). In summary, MCT1 expression levels were not associated with patient age, tumor size, tumor tissue types, or tumor invasiveness.

### 3.10. Correlation Between BACH1 and MCT1 Expression in Breast Tumors

Our recent study revealed that BACH1 acts as a transcriptional suppressor of *SLC16A1*, which encodes MCT1, suppressing lactate catabolism in TNBC cells (17). We therefore investigated whether BACH1 and MCT1 expression levels were correlated in our patient TMA cohort, where we collected both BACH1 and MCT1 IHC scores. Using Spearman’s rank correlation analyses, we observed a positive correlation (0.376, *p* < 0.001) between BACH1 and MCT1 in total breast tumors (*N* = 114) (Figure 6A). When we further analyzed their correlation in each tumor subtype, we similarly observed a positive but insignificant trend between BACH1 and MCT1 in the basal-like (0.226, *p =* 0.222; *N =* 31), luminal A (0.237, *p =* 0.048; *N* = 70), and luminal B (0.353, *p* = 492; *N* = 6) subtypes (Figure 6B). In contrast, we noticed an inverse, but not significant, correlation between BACH1 and MCT1 IHC scores in the HER2^+^ subtypes (−0.17) from a smaller sample (*N* = 7). Interestingly, the expression correlation between BACH1 and MCT1 differed noticeably by patient race. Among Black patients, BACH1 and MCT1 displayed a strong positive correlation for their expression (r = 0.525, *p* < 0.00001). In tumors among White women, however, there was a relatively weak and insignificant Spearman’s rank correlation between BACH1 and MCT1 expression (r = 0.186, *p* = 0.211), indicating racial disparity between BACH1 and MCT (Figure 6C). Taken together, our data demonstrate a strong expression correlation between BACH1 and MCT1 in breast tumors from Black women, but not in those from White women, and mostly from the luminal A breast tumor subtype.

## 4. Discussion

The heme-regulatory transcriptional factor BACH1 promotes the epithelial to mesenchymal transition (EMT), invasiveness, and migration of cancer cells, as well as the metastasis of breast tumors. BACH1 also regulates metabolic networks of cancer cells to support breast cancer metastasis. Therefore, tumoral BACH1 expression predicts a worse outcome for patients with breast cancer. These data support our initiatives to investigate BACH1 protein levels in breast tumors from individual patients with various biological and racial backgrounds to identify any clinical relevance as an indicator.

In the current study, we observed that BACH1 expression levels are positively correlated with tumor size and are significantly higher in histological Grade 3 tumors. These observations suggest that BACH1 expression might be increased as tumors rapidly proliferate, and/or BACH1 might enhance its protein stability or transactivation in the tumor microenvironment because of tumor extrinsic factors as tumors grow, which needs further investigation. Indeed, BACH1 mRNA expression is induced by hypoxia, which implies a tumor microenvironment effect [36]. Oxygen tension or diffused oxygen concentration in the tumor microenvironment might directly or indirectly affect BACH1 protein expression levels to promote cancer metastasis [3]. A surprising finding in our study was that BACH1 expression in breast tumors differs by race. Specifically, BACH1 expression levels are higher in basal-like subtype tumors than in other subtypes in Black women. In contrast, BACH1 expression levels did not show statistical differences in basal-like tumors or other subtypes in White women. These results could be linked to differences in tumor sizes between the groups. Black women experience larger tumors, and we found that BACH1 levels correlated with tumor size. Our data suggested that BACH1 levels were particularly abundant in basal-like breast tumors from Black women, highlighting BACH1 as a distinct patient race-related biomarker. It also suggests the need for further study to decipher the genomic stability or amplification of BACH1 in tumors from Black women. Our BACH1 IHC analyses, along with complementary gene expression data from breast cancer patients, suggest that BACH1 is a distinct protein for TNBC [37].

Akin to BACH1, MCT1 is a known prognosis marker for breast cancer [18,19], and the inhibition of MCT1 using AZD3965 is currently under investigation in clinical trials to be applied as a targeted cancer therapy [22]. It has previously been published that breast tumors expressed elevated MCT1 levels compared to adjacent normal tissue [19,21,22]. In our current study, both Black and White women showed that MCT1 expression levels were significantly elevated in the basal-like tumors when compared with other subtypes, suggesting MCT1 as a useful biomarker in patients with basal-like tumors to predict the efficacy of AZD3965 treatment. Moreover, we found that there is no appreciable difference or association between MCT1 expression and either tumor size, tumor tissue type, or patient age. Likewise, BACH1 showed no statistical differences among patients grouped by tissue type or patient age. These data strongly suggest that the aging of breast cancer patients is less likely to affect the levels of BACH1 or MCT1 in tumors. Furthermore, other clinical variances, including the inflammation status and metabolic disease status, such as obesity or type II diabetes of patients, might be of interest in future research on BACH1 or MCT1 expression.

Of note, there was a positive correlation between BACH1 and MCT1 expression using IHC scores, particularly among Black women. Since our previous study demonstrated that BACH1 suppresses MCT1-dependent lactate oxidation pathways in TNBC [17], a negative correlation was expected between BACH1 and MCT1 expression in the basal-like subtype of breast tumors. Small sample size would not provide enough power to compare the expression in basal-like or HER2^+^ subtypes of tumors, which is noted as a limitation of our current study, although the study included patient race, tumor subtypes, and clinical variables. In particular, it should be noted that consideration of BACH1 expression levels, particularly in luminal B subtype tumors among Black and White women, using a larger patient dataset would provide a better understanding of whether BACH1 expression is dependent on race or tumor subtypes. In addition, it is suggested that additional analyses of MCT1 and BACH1 expression at the single-cell level in each tumor type be conducted, as we analyzed total IHC scores per tumor slide rather than examining cellular levels of each molecule. This method could address and advance the limitation of our current analysis, as correlation analyses using tumor tissues containing heterogeneous cancer cell populations cannot access the possibility of exclusive expression between BACH1 and MCT1 at the single-cell level. Furthermore, the inclusion of other known predictive markers in the future study would support our findings, suggesting that, together with BACH1, they may serve as a stronger indicator of patient outcome.

## 5. Conclusions

In our comprehensive analyses of BACH1 and MCT1 expression in breast tumors using IHC assays, we observed substantial increased expression levels of BACH1 in the tumors from Black patients, particularly in the basal-like breast tumors. BACH1 expression shows a positive correlation with tumor size, whereas MCT1 expression was not significantly correlated with tumor size. Higher tumor grade (grade 3) correlated with the highest levels of BACH1 and MCT1 in our patient cohort. A positive association between BACH1 and MCT1 was detected in breast tumors among Black women, whereas a null association was found in tumors among White women. Our data suggest BACH1 as a distinct race-associated biomarker for breast cancer.

## Figures and Tables

**Figure 1 curroncol-32-00404-f001:**
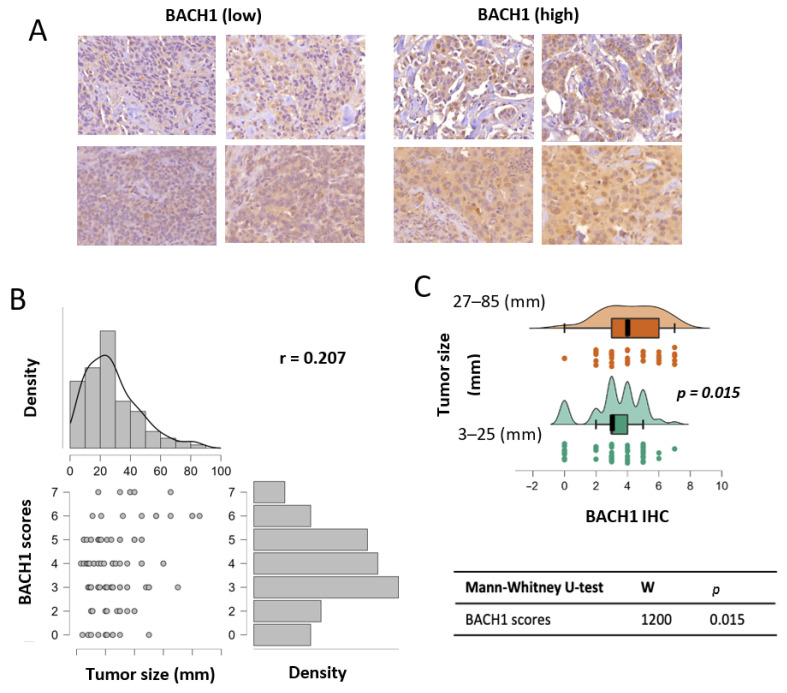
BACH1 expression is positively associated with breast tumor size. (**A**) Representative images of BACH1 staining via IHC assays using breast tumor tissues. Staining scores are displayed from low to high. (**B**) BACH1 IHC score correlation with tumor size (diameter). Total sample *N* = 115; *Spearman’s* correlation coefficient = 0.207 (*p* = 0.027). (**C**) Patient tumors were separated into two groups based on tumor sizes (3–25 mm in diameter, *N* = 65) and (27–85 mm in diameter, *N* = 50). This split corresponded to the 50th percentile of the tumor size distribution. Distribution plot showing BACH1 IHC scores divided into two tumor size groups via the Mann-Whitney U test.

**Figure 2 curroncol-32-00404-f002:**
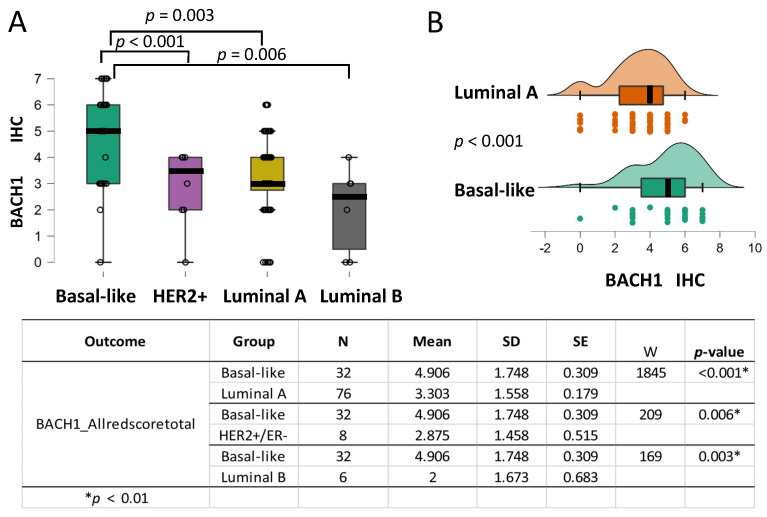
BACH1 expression is higher in the basal-like subtype than in the other breast tumor subtypes. (**A**) Box plots indicating BACH1 IHC scores by breast cancer subtype: basal-like (*N* = 32), HER2^+^ positive (*N* = 7), luminal A (*N* = 76), and luminal B (*N* = 5) subtypes. (**B**) Independent group comparison of BACH1 scores in basal-like (*N* = 32) and luminal A (*N* = 76) subtypes showing significantly higher BACH1 scores in basal-like tumors than in the luminal A subtype (*p* < 0.001, Rank-Biserial Correlation = 0.517), according to a Mann–Whitney U-test for cross-comparison. Score mean, standard deviation (SD), standard error (SE), and coefficient of variation in the subtype groups are shown by the subtype groups. The same trend is observed when comparing the basal-like subtype with the HER2^+^ subtype (*p* = 0.006) and the basal-like subtype with the luminal B subtype (*p* = 0.003).

**Figure 3 curroncol-32-00404-f003:**
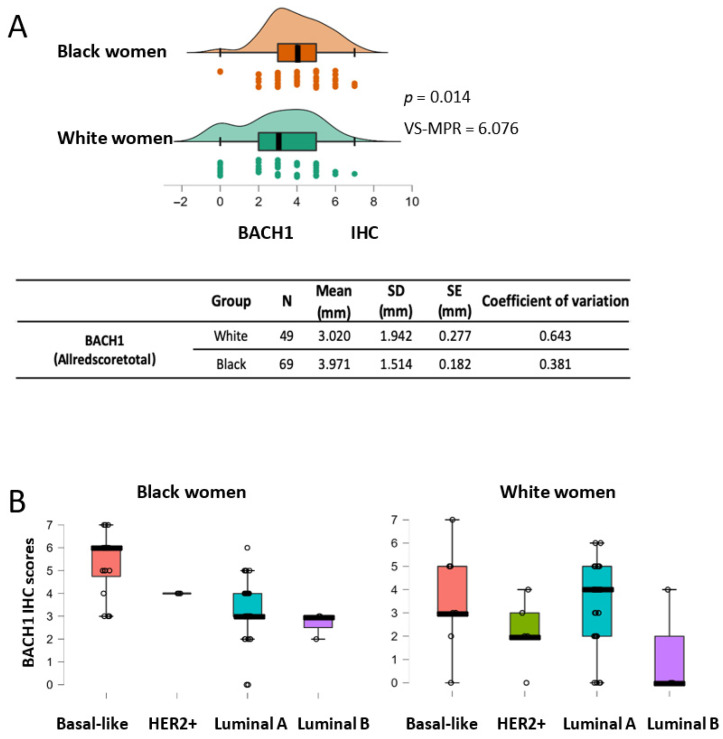
BACH1 expression scores are higher in Black women than in White women, as well as in basal-like subtype tumors from Black women. (**A**) BACH1 IHC score comparison between Black women (*N* = 69) and White women (*N* = 49) using the Mann–Whitney U test (*p* = 0.014, VS-MPR is 6.076). Mean score of tumor size (millimeter, mm), standard deviation (SD), standard error (SE), and coefficient of variation are shown by ethnic group. (**B**) Box plots indicate BACH1 IHC scores by tumor subtypes among Black or White patients.

**Figure 4 curroncol-32-00404-f004:**
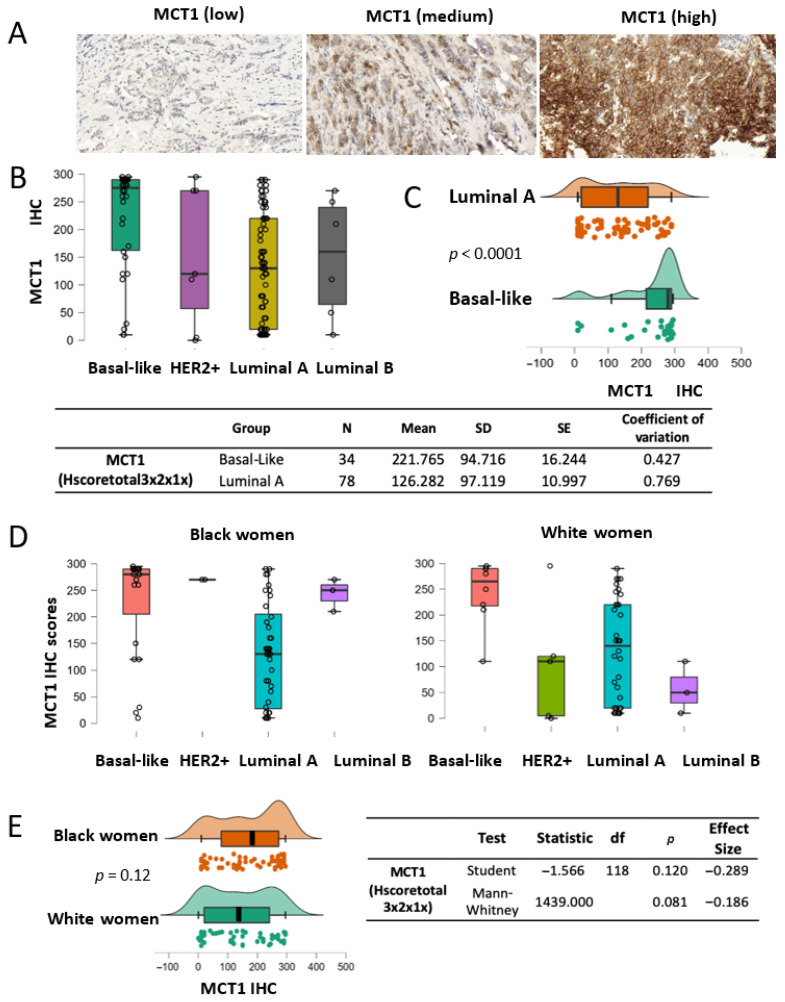
MCT1 expression by breast cancer subtype. (**A**) Representative IHC images of MCT1 staining with low to high intensity in the breast TMA. (**B**) Boxplots showing MCT1 IHC scores by the subtypes of breast tumors (basal-like *N* = 31, HER2^+^ *N* = 7, luminal A *N* = 70, and luminal B *N* = 6). (**C**) MCT1 score analysis between luminal A (*N* = 78) and basal-like subtypes (*N* = 34) (*p* < 0.0001) using the Mann–Whitney *U* test. An effect size of 0.561 is given by the rank-biserial correlation. (**D**) Boxplots showing MCT1 IHC scores by tumor subtype from Black (**left**) and White women (**right**). (**E**) Distribution plots indicating that MCT1 scores have no statistical difference between race groups according to Student’s *t* test (*p* = 0.12) or the Mann–Whitney U test (*p* = 0.081).

**Figure 5 curroncol-32-00404-f005:**
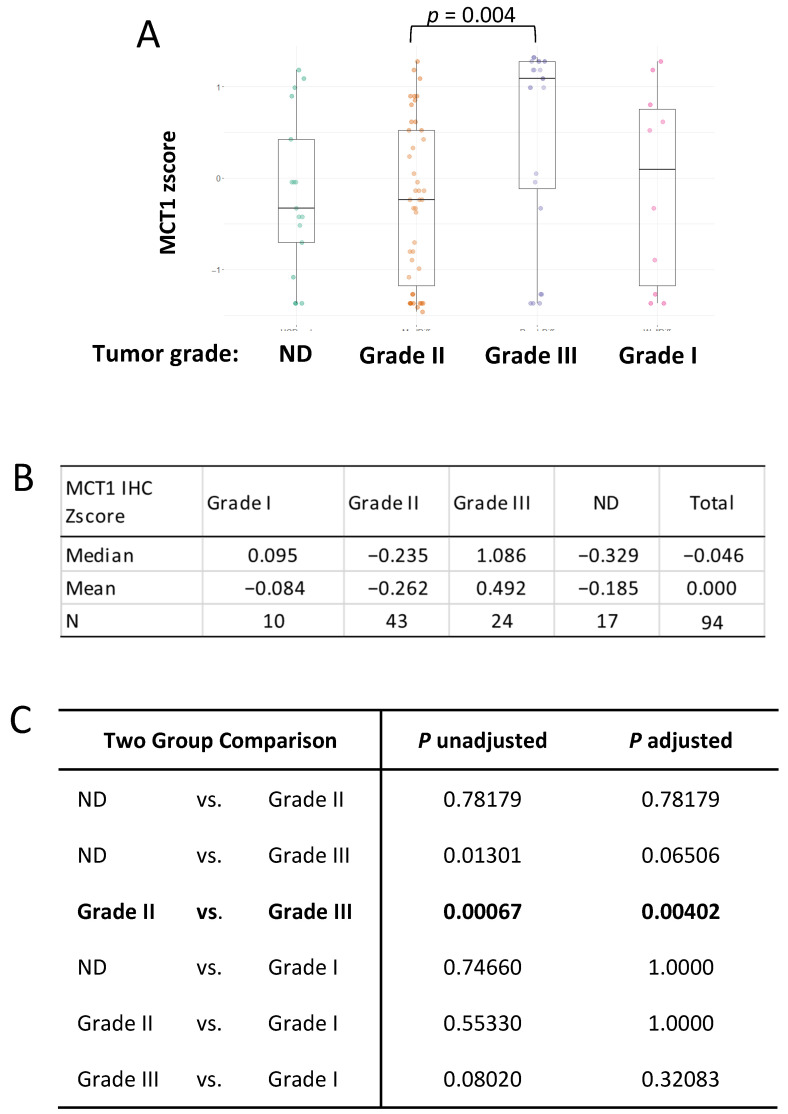
MCT1 expression is higher in the Grade 3 tumor subgroup. (**A**) Box plots indicating MCT1 IHC z-scores by breast cancer grade using the Kruskal–Wallis’s test. (**B**) Median and mean of MCT1 IHC z-scores in each group; Grade 1 (*N* = 10), Grade 2 (*N* = 43), Grade 3 (*N* = 24), and Not Determined (*N* = 17) are shown. (**C**) Kruskal–Wallis’s multiple-comparison, with *p*-values adjusted with the Holm method, is shown in the table.

**Figure 6 curroncol-32-00404-f006:**
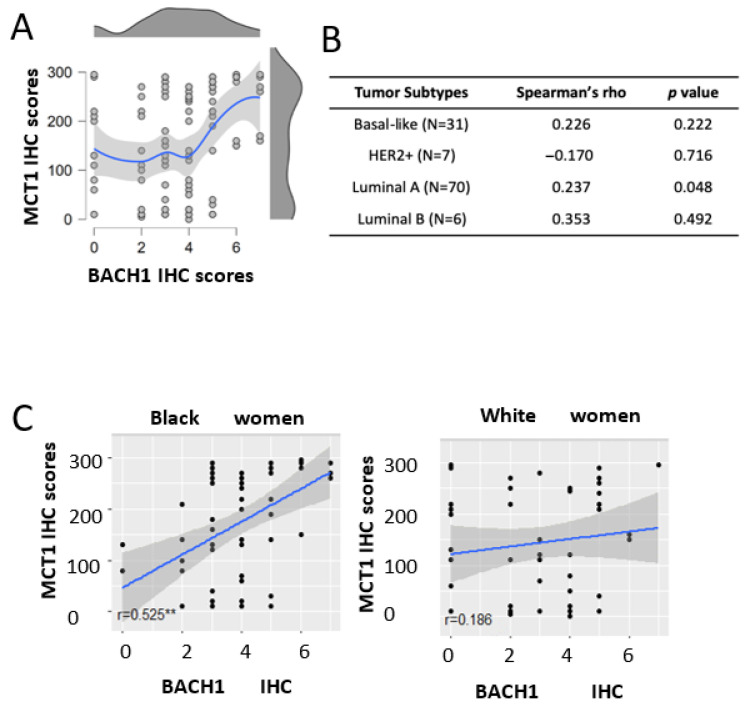
Expression analysis of MCT1 and BACH1 in breast tumors. (**A**) Scattered plot and distribution plot of BACH1 and MCT1 scores show a positive expression correlation according to Spearman’s rank correlation analysis (0.376, *p* < 0.001). The red dot in the distribution plot represents the median of the MCT1 variable at each level of BACH1. (**B**) Correlation analyses between BACH1 and MCT1 in tumor subtypes (patient numbers). *Spearman’s rho* and *p*-values are shown. (**C**) Scatter plot comparing the association between BACH1 and MCT1 by race: Black women (*N* = 62, *Spearman’s Correlation* rho = 0.525, *p* = 0.00001, *N* = 62) vs. White women (*Spearman Correlation* rho = 0.186, *p* = 0.211, *N* = 47), ** Significant.

## Data Availability

The original contributions presented in this study are included in the article/Appendix A. Further inquiries can be directed to the corresponding author.

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
