# Peer review of "Differential BACH1 Expression in Basal-like Breast Tumors of Black Women Identified via Immunohistochemistry"

_curroncol, 2025, doi:10.3390/curroncol32070404_

Round 1

Reviewer 1 Report

Comments and Suggestions for Authors

The manuscript described a potential biomarker to predict breast cancer outcome with only two biomarkers. Other biomarkers could also predict to predict the outcome. These two markers may not fully predict the outcome. the manuscript should include other convincing biomarkers that have been well studied.

Try to identify predictive markers for breast cancer, but this study requires to  include other known predictive markers to support their findings.

require review the publications of other predictive markers  

Add other references for other predictive markers

Comments on the Quality of English Language

NA

Author Response

Comment 1: The manuscript described a potential biomarker to predict breast cancer outcome with only two biomarkers. Other biomarkers could also predict to predict the outcome. These two markers may not fully predict the outcome. the manuscript should include other convincing biomarkers that have been well studied.

Response 1: Thank you for the comment. and I agree with the reviewer that relying on two biomarkers may not fully capture the complexity of predicting patient outcomes. However, I would like to reiterate that the primary objective of this study is to validate BACH1 as a prognostic biomarker using breast tumor tissue samples. Our focus is on evaluating and highlighting the clinical significance of BACH1 as a valuable indicator of prognosis, rather than direct comparison with other biomarkers, which may also be important. Notably, we observed that BACH1 expression is elevated in basal-like and large tumors from Black patients—a finding that, to our knowledge, has not previously evaluated in relation to patient race. Therefore, we aimed to determine whether BACH1, along with its downstream target MCT1, provides additionally prognostic information. It is worth mentioning that MCT1 has been previously reported as breast cancer biomarker (Hong et al. 2016. Cell Reports, Citation #18), and thus could serve be an complementary known biomarker in our study.

Comment 2: Try to identify predictive markers for breast cancer, but this study requires to  include other known predictive markers to support their findings.

Response 2:  I appreciate the reviewer’s comment. However, we believe that it is challenging to direct support BACH1’s value as a biomarker with well-known markers such as EGFR, VEGF, TP53, PI3KCA, BRCA1/2 as they are largely independent in their biological roles and mechanisms. Nonetheless, we acknowledged the reviewer’s points and have incorporated it into the Discussion section as a limitation of the current study and highlights the potential need for future studies to explore combinatorial or comparative biomarker strategies.

Comment 3: require review the publications of other predictive markers  

Response 3: Thanks to the reviewer’s comment, we added other predictive markers and included in the manuscript.

Comment 4: Add other references for other predictive markers

Response 4: Thanks to the reviewer’s comment, we have included references for other predictive markers in the manuscript.

Reviewer 2 Report

Comments and Suggestions for Authors

This study systematically evaluate BACH1 protein expression in human breast tumor using immunohistochemistry and innovatively investigate its association with race-related disparities.

Comments:

  1. Are specific combinations of BACH1 and MCT1 expression levels (e.g., both positive, one positive/one negative, or both negative) associated with clinicopathological characteristics?
  2. Survival analyses based on expression levels of BACH1 and MCT1 could be added if survival data are available.
  3. Please ensure that the p-values in Figure 2 are clearly and accurately labeled.
  4. Please consider adopting a consistent and unified color scheme across all figures throughout the manuscript.

The manuscript is suitable for publication pending minor revisions.

Author Response

Comments:

  1. Are specific combinations of BACH1 and MCT1 expression levels (e.g., both positive, one positive/one negative, or both negative) associated with clinicopathological characteristics?

Response: We revealed that BACH1 regulates metabolic pathways, including lactate catabolism in TNBC cells, by suppressing monocarboxylate transporter 1 (MCT1 or encoded by SLC16A1) that facilitates pyruvate and lactate transport via a proton-dependent mechanism (15–17). Also, MCT1 has been previously reported as breast cancer biomarker (18). Thus this study is to validate BACH1 as a prognostic biomarker using breast tumor tissue samples.

  1. Survival analyses based on expression levels of BACH1 and MCT1 could be added if survival data are available.

Response: Thanks to the reviewer’s invaluable comment. We are currently collecting the data; however, it is not yet sufficient for survival analysis.

  1. Please ensure that the p-values in Figure 2 are clearly and accurately labeled.

Response: Thanks to the reviewer’s comment, we confirmed p values in Figure 2.

  1. Please consider adopting a consistent and unified color scheme across all figures throughout the manuscript.

Response: Thanks to the reviewer’s comment, we modified color schemes but try to differentiate data plots of BACH1 and MCT1 per subtypes and size using different colors and used consistent same color code per race for Figure 2A, 3B, 4B, 4D. For distribution of BACH1 or MCT1, we used 2 colors (orange or green) for Figures 1C, 2B, 3A, 4C, 4E. If figures need further modification, we would do it.

Reviewer 3 Report

Comments and Suggestions for Authors

Dear authors,

Thank you for the opportunity to review your paper. The need for predictive biomarkeri in Breast cancer are very much need. I have some coment about your manuscript:

  • this a retrospective study- the period of retrospective analises  have to be clearly descried.
    - the description of patient population included in the study is also unclear. You have to describe patient population.
  • You have resuts also related to two biomarkers: BACH1 and MCT1 but the second marker is not in the title and I think that it should be.
  • your study is relatated about relationship between BACH1 and MCT1 expression and epidemiological, clinical and pathological characteristics of breast cancer patients. And you have no data about outcomes of this patients. So you have to change accordingly the title and the obiective of the study.

Author Response

Thank you for the opportunity to review your paper. The need for predictive biomarkeri in Breast cancer are very much need. I have some coment about your manuscript:

  • this a retrospective study- the period of retrospective analises  have to be clearly descried. 
    - the description of patient population included in the study is also unclear. You have to describe patient population.

Response: Thanks to the reviewer’s precise comment. The period of retrospective study has been included in the Materials and Method section. And the patient information has been provided in Table 1 and 2 in the manuscript.

  • You have resuts also related to two biomarkers: BACH1 and MCT1 but the second marker is not in the title and I think that it should be.

Response: Thanks to the reviewer’s comment. The primary objective of this study is to validate BACH1 as a prognostic biomarker using breast tumor tissue samples. Our recent study revealed that BACH1 regulates metabolic pathways, by suppressing MCT1. However, MCT1 has been previously reported as breast cancer biomarker by other group (18).

  • your study is relatated about relationship between BACH1 and MCT1 expression and epidemiological, clinical and pathological characteristics of breast cancer patients. And you have no data about outcomes of this patients. So you have to change accordingly the title and the obiective of the study.

Response: Thanks to the reviewer’s comment. We are currently collecting the data; it is not yet sufficient for survival analysis. However, we modified the title to “ BACH1 expression is a race-related distinct predictor for triple-negative breast cancer patient.”